# Abrupt Dietary Change and Gradual Dietary Transition Impact Diarrheal Symptoms, Fecal Fermentation Characteristics, Microbiota, and Metabolic Profile in Healthy Puppies

**DOI:** 10.3390/ani13081300

**Published:** 2023-04-11

**Authors:** Pinfeng Liao, Kang Yang, Hongcan Huang, Zhongquan Xin, Shiyan Jian, Chaoyu Wen, Shansong He, Lingna Zhang, Baichuan Deng

**Affiliations:** Guangdong Provincial Key Laboratory of Animal Nutrition Control, National Engineering Research Center for Breeding Swine Industry, College of Animal Science, South China Agricultural University, Guangzhou 510642, China

**Keywords:** dietary change, beagle dog, fermentation characteristics, fecal microbiota, metabolomics

## Abstract

**Simple Summary:**

In this study, dietary changes in puppies were observed to cause different gastrointestinal responses. Using two change methods, one direct and one gradual, we found that a gradual transition reduced the incidence of diarrhea in puppies throughout the trial period, as well as the concentration of isovaleric acid. Meanwhile, 16S rRNA sequencing showed that the fecal microbiota was changed after different dietary changes. Compared with the bacterial changes after an abrupt dietary change, the relative abundances of beneficial bacteria (i.e., *Turicibacter* and *Faecalibacterium*) in feces were increased after a gradual dietary transition in puppies. Additionally, both change methods caused changes in amino acid metabolism, while an abrupt change also altered lipid metabolism. An abrupt change increased fecal histamine and spermine concentrations, but decreased concentrations of metabolites such as 5-hydroxyindoleacetic acid and serotonin. Our findings indicated that a gradual transition most likely reduced the diarrhea rate in puppies by modulating the composition and metabolism of the gut microbiota.

**Abstract:**

Dietary changes are inevitable for pets, yet little is known about the impact of different dietary change methods on the gastrointestinal response. The current comparative study evaluated the effects of different dietary changes on the diarrheal symptoms, fecal fermentation characteristics, microbiota, and metabolic profile of healthy puppies. A total of 13 beagle puppies were randomly divided into two groups; puppies in the abrupt change (AC) group were given 260 g of a chicken- and duck-based extruded diet (CD)daily for the one-week transition period, whereas puppies in the gradual transition (GT) group were fed according to a gradual transition ratio of a salmon-based extruded diet (SA) and a CD diets with a difference of 40 g per day for seven consecutive days. Serum samples were collected on D7, and fecal samples were collected on D0 and D7. The results indicated that GT reduced the incidence of diarrhea in puppies throughout the trial period. Dietary change methods had no influence on serum inflammatory factors or fecal SCFAs, but isovaleric acid was significantly reduced after GT. Meanwhile, 16S rRNA sequencing showed that the fecal microbiota was changed after different dietary changes. Compared with the bacterial changes after AC, the relative abundances of beneficial bacteria (i.e., *Turicibacter* and *Faecalibacterium*) in feces were increased after GT in puppies. Additionally, both GT and AC caused changes in amino acid metabolism, while AC also altered lipid metabolism. AC increased fecal histamine and spermine concentrations, but decreased concentrations of metabolites such as 5-hydroxyindoleacetic acid and serotonin. Our findings indicated that GT most likely reduced the diarrhea rate in puppies by modulating the composition and metabolism of the gut microbiota.

## 1. Introduction

It is well-known that feeding different diets to dogs and cats at different life stages is necessary to meet their nutrient requirements [1]. In other species, dietary changes often induce changes in nutrient digestion, absorption, growth performance, and the gut microbiota (GM) due to different dietary compositions [2,3,4,5]. A sudden dietary change often results in diarrhea in pets, but the underlying mechanism remains unknown [6]. Diarrhea is highly associated with gut microbiota alterations. Moon et al. determined colonization of the small intestine as initial evidence pointing to the notion that *Escherichia coli* caused diarrheal disease in newborn pigs now recognized as enterotoxigenic E. coli infection [7,8]. Similarly, *Shigella*, *Salmonella*, and *Clostridium difficile* cause diarrhea, and toxic substances produced by these pathogens further cause abnormal gut function and immune responses, leading to the occurrence of diarrhea [8,9]. At the same time, harmful bacteria can compete for resources and implant sites in the gastrointestinal tract, thus inhibiting the number of beneficial bacteria [8,9,10,11]. Conversely, beneficial bacteria such as *Lactobacillus*, yeast, and *Bifidobacterium* can be applied to treat pathogen-caused diarrhea by maintaining the balance of GM [12]. Other beneficial bacteria, such as *Firmicutes*, *Faecalibacterium*, and *Coprococcus* bacteria, produce short-chain fatty acids (SCFAs), which serve as the carbon energy for intestinal epithelial cells, and SCFAs help to maintain the colorectum function, as well as the morphology and function of colonic epithelial cells [13]. Recent evidence has begun to link the gut microbiome and its metabolites in mice and humans to gastrointestinal diseases and inflammation [14,15]. Therefore, we hypothesized that the diarrhea problems caused by sudden dietary changes may be caused by changes in the GM.

The current study investigated and compared the effects of two dietary change protocols, namely abrupt change (AC) and gradual transition (GT), on puppies. We evaluated the stool quality, incidence of diarrhea, inflammatory responses, and fecal SCFA content at the beginning and end of the dietary change period. Meanwhile, 16Sr RNA gene sequencing and untargeted metabolomics were adopted to capture changes in the microbiota and metabolic pathways, as well as to identify potential metabolic matters. Therefore, the objective of this research was to determine whether there were key bacteria and/or metabolites to expound the influence of dietary changes on puppies, and whether GT could reduce the changes in GM and metabolism attributed to dietary changes, as well as maintain health in puppies.

## 2. Materials and Methods

### 2.1. Animal, Diet, Treatment, and Experimental Design

All processes were approved by the Animal Care and Use Committee prior to animal experimentation (Approval number: 2019188), following the principles of the Center of Laboratory Animal at South China Agricultural University. Animal care staff monitored animal health on a daily basis.

Thirteen 6-month-old beagle puppies were housed individually in a metabolic cage (1.2 m × 1.0 m × 1.1 m kennels) under a relatively constant environment, with a humidity of 70% ± 5%, temperature of 23 ± 1 °C, and 12 h dark/light cycle at the Center of Laboratory Animal of South China Agricultural University. Starting from 1 month before the experiment, none of the animals were dewormed or given drugs that could alter the GM, e.g., antibiotics. All puppies had access to toys at all times and were allowed to socialize outside their cages with humans and other animals for about 1 h at least 3 days a week. Clean water was freely available, and food was offered twice daily throughout the trial.

Two commercial extruded diets, a salmon-based extruded diet (SA) and a chicken- and duck-based extruded diet (CD), were purchased from Foshan Ramical Animal Nutrition and Health Care Technology Co., Ltd. (Foshan, China). Both diets were made from similar ingredients, including corn flour, flour, fish oil, chicken meal, duck meal, beef meal, fish meal, soybean meal, and amino acid, vitamin, and mineral premixes. The chemical and energy compositions of the basal diets are presented in Table 1. Both diets complied with all nutritional recommendations for puppies by the American Association of Feed Control Officials (AAFCO, 2017). According to the National Research Council (NRC, 2006), each dog had a total feed intake of 260 g, split into two meals per day.

These puppies first adapted to the SA diet for two months, and then were changeed to the CD diet. An outline of the dietary transition and the timeline of events is shown in Figure 1. All puppies were weighed and randomly divided into two groups at D0; there were no differences in body weight (BW) or BCS between the two groups. Puppies in the AC group (*n* = 7, four females and three males) were given 260 g of the CD diet daily for the one-week transition period. Puppies in the GT group (*n* = 6, four females and two males) were fed a gradual transition ratio of SA and CD diets for seven consecutive days as follows: 260:0, 220:40, 180:80, 140:120, 120:140, 80:180, 40:220, and 0:260 g. Body condition scores (BCS) were performed for each beagle on D0 and D7, using the Laflamme method [16]. Fecal scores (FS) were performed for each beagle on D1–D7, using the Middelbos method [17]. A soft stool rate refers to 3.5 ≤ FS < 4; a diarrhea rate refers to 4 ≤ FS < 5.

### 2.2. Chemical Analysis of Diet

The SA and CD diets were collected on D1 and D7 of the dietary transition, and then stored in a dry pot. The feed samples were dried in an oven and pulverized, and then passed through a 1 mm screen for chemical composition analysis. The chemical and energy compositions of SA and CD diets are shown in Table 1.

### 2.3. Fresh Fecal Sample Collection and Preparation

A clean tray was placed under each dog cage to obtain fresh feces. During the two-week period, fecal samples were scored daily according to Middelbos [17], and fresh feces were collected within 15 min on D0 and D7 of the dietary transition. The feces from one animal were divided into three samples. Among them, two fecal samples were packed into 1.5 mL sterile and enzyme-free EP tubes for metabolomic analysis and measurement of SCFAs and branched-chain fatty acids (BCFAs), whereas one fecal sample was packed into a 5 mL germfree fecal collection tube for microbiota measurement. All samples were snap-frozen using liquid nitrogen and then transferred to a −80 °C refrigerator for storage until analysis. The fecal samples were prepared for SCFA and BCFA analysis according to previous research in our laboratory [18]. Briefly, 0.2 g of each fecal sample in a 2 mL tube was combined with 1 mL of ultrapure water, and then vortexed for 2 min. The samples were subjected to ultrasonic crushing at 4 °C for 10 min and centrifuged at 13,000 rpm, 4 °C for 10 min. The supernatant was transferred into a new 2 mL centrifuge tube, before adding 20 μL of 25% metaphosphoric acid and 0.25 g of anhydrous sodium sulfate. The mix was then vortexed for 1 min, followed by the addition methyl tert-butyl ether in constant volume to 2 mL in the fuming cupboard. After vortexing for 5 min, the tubes were centrifuged at 13,000 rpm, 4 °C for 5 min. The supernatant was filtered using a 0.22 μm membrane for GC–MS/MS analysis.

### 2.4. Blood Sample Collection and Analysis

A 5 mL blood sample was collected via the forelimb vein to determine the serum levels of immune factors on D7. The collected blood was left standing for 30 min and centrifuged at 3000 rpm at 4 °C for 15 min. The supernatant of each sample was evenly distributed into three Eppendorf tubes and stored at −80 °C. The levels of interferon-γ (IFN-γ, MM-35063O1), tumor necrosis factor-alpha (TNF-α, MM-36988O1), interleukin-4 (IL-4, MM-35084O1), interleukin-2 (IL-2, MM-85058O1), interleukin-6 (IL-6, MM-1546O1), immunoglobulin A (IgA, MM-85082O1), immunoglobulin M (IgM, MM-85090O1), and immunoglobulin G (IgG, MM-2086O1) in the serum of beagle puppies were determined by enzyme-linked immunosorbent assay (ELISA) using commercial kits (MEIMIAN, Jiangsu, China) following the products’ instructions.

### 2.5. Extraction of DNA and High-Throughput Sequencing

Total bacteria DNA was extracted from frozen fecal samples with the Stool DNA Kit following the standard protocol (Tiangen, Beijing, China). The DNA concentration and purity were examined by 1% agarose gel electrophoresis. Then, DNA was diluted to 1 ng/μL in germfree water.

Quantitative insights into microbial ecology were used to analyze the sequencing data bioinformatics. The 16S V3–V4 rRNA was amplified with region-specific primers (i.e., 341F: CCTAYGGGRBGCASCAG and 806R: GGACTACNNGGGTATCTAAT), where F and R denote forward and reverse, respectively. The PCR reaction was performed using 15 μL of Phusion^®^ High-Fidelity PCR Master Mix (New England Biolabs, Ipswich, MA, USA) with the following conditions: 98 °C for 1 min (1 cycle) for initial denaturation, followed by 30 cycles at 98 °C for 10 s for denaturation, 50 °C for 30 s for annealing, 72 °C for 30 s for elongation, and a last step of 72 °C for 5 min for final extension. The same volume of 1× loading buffer was mixed with PCR products according to equidensity ratios, and the mixture of PCR products was purified using the Qiagen gel extraction kit (Qiagen, Hilden, Germany). Sequences with main band sizes between 400 and 500 bp were selected. The paired-end sequencing was executed by the Illumina NovaSeq 6000 platform (Novogene, Tianjin, China) according to the standard protocol from the manufacturer.

### 2.6. Bioinformatics Analysis

The bioinformatics analysis of sequencing data was carried out through Quantitative Insights into Microbial Ecology (QIIME, V1.9.1, http://qiime.org/scripts/split_libraries_fastq.html, accessed on 2 March 2022). Paired-end sequences from the original DNA-overlapped fragment were merged using FLASH (V1.2.7, http://ccb.jhu.edu/software/FLASH/, accessed on 2 March 2022). Sequence demultiplexing and stitching, quality filtering, and analysis were performed using the unweighted pair-group method with arithmetic means (UCHIME, http://www.drive5.com/usearch/manual/uchime algo.html, accessed on 2 March 2022), and the effective tags were obtained.

Operational taxonomic units (OTUs) with a 97% similarity threshold in the sequence were subsequently normalized in order to analyze α and β diversities. The sequence was chosen as a representative for each OTU, and the RDP Classifier (V2.2, http://sourceforge.net/projects/rdp-classifier/, accessed on 2 March 2022) was used to label taxonomic information for each representative sequence. A representative sequence was picked for each OUT, and taxonomic information was annotated using the mothur algorithm by the Silva Database (http://www.arb-silva.de/, accessed on 2 March 2022). MUSCLE software (V3.8.31, http://www.drive5.com/muscle/, accessed on 2 March 2022) was used to carry out multiple-sequence alignment to study the phylogenetic relationship of different OTUs. The anosim index and α diversity (i.e., observed species, Chao1, Shannon, Simpson, ACE, and PD_whole_tree) for the significance of differences within and between groups and the complexity of species diversity for an individual sample were generated with QIIME (V1.7.0) and displayed with R software (V2.15.3). The β diversity for evaluating differences among samples in species complexity was calculated by QIIME software. Cluster analyses containing weighted_unifrac and unweighted_unifrac distances were performed using principal coordinate analysis (PCoA) and displayed using the WGCNA package, stat package, and ggplot2 package in R software (V2.15.3). Linear discriminant analysis coupled with effect size (LEfSe) was adopted to distinguish the bacterial taxa differentially represented between groups at genus or higher taxonomy levels. The default setting of LEfSe software was an LDA score of >4 (http://huttenhower.sph.harvard.edu/lefse/, accessed on 25 June 2022).

### 2.7. Metabolite Extraction

The feces were also used for metabolomics analysis, and the sample preparation began with 60 mg of fecal sample in a 2 mL tube, to which 600 μL of methanol/water (1:1, *v*/*v*) and 20 μL internal standard (L-2-chlorophenylalanine, 0.3 mg/mL, methanol configuration) were added. Sample tubes were homogenized and exposed to 10 min of ultrasonic crushing. Samples were left standing at −20 °C for 30 min. Centrifugation was then performed at 14,500 rpm, 4 °C for 15 min; then, 200 μL of supernatant was transferred into a new tube and steamed to dry by a vacuum centrifuge. Next, 200 μL of methanol/water (1:1, *v*/*v*) was added to each tube. After vortexing for 30 s, the tubes were exposed to ultrasonic crushing at 4 °C for 10 min and centrifuged at 14,500 rpm, 4 °C for 15 min. Finally, all supernatant was filtered using a 0.22 μm membrane for LC–MS/MS analysis. Quality control (QC) was performed by mixing different individual fecal samples at each into a 2 mL tube, referring to the above process.

### 2.8. UPLC–Orbitrap–MS/MS and Metabolite Profiling Analysis

The process of UPLC–Orbitrap–MS/MS analysis was adapted from [19]. Analytical instruments usually cannot provide undefiled and visualized information about metabolites. Sequence preprocessing of the original data is needed to obtain a feasible data matrix, and includes noise filtering and baseline correction, peak detection and deconvolution, alignment, and normalization. Compound Discoverer 2.1 (CD, Thermo Fisher Scientific), a flexible and automated data analysis tool, was used to perform data analysis. CD can identify small molecular metabolites with high accuracy using various tools including mzCloud (online spectral library with >2 million spectra), ChemSpider (chemical structure database with >500 data sources, 58 million structures), mzVault (local spectral libraries), and Masslist (local databases).

### 2.9. Statistical Analysis

All statistical computation was conducted using SPSS 26.0 (IBM, Amonk, NY, USA). Significances and tendencies were set at *p* < 0.05 and 0.05 ≤ *p* < 0.10, respectively. Paired or unpaired Student’s *t*-tests and chi-square tests were conducted to assess the main treatment effect on different measures, and results were visualized using GraphPad Prism 8.0.2 (GraphPad Software Inc., La Jolla, CA, USA). Data are expressed as the means ± standard error of the mean (SEM) unless otherwise stated in the text and legends.

Metaboanalyst 5.0 (https://www.metaboanalyst.ca/, accessed on 15 October 2022) was used to perform principal component analysis (PCA). Partial least-squares discriminant analysis (PLS-DA) and pathway impact analysis were also performed in this study. The results were visualized with Metaboanalyst 5.0. Spearman’s Rho was performed to calculate the correlation between the relative abundance of different microbiota and the relative abundance of metabolites.

## 3. Results

### 3.1. FS, Diarrhea Rate

No abnormal feeding behavior, such as changes in dietary intake, or other health problems were observed in the puppies during the dietary transition period. Interestingly, the AC group had significantly higher mean FS than the GT group during the dietary transition period (*p* < 0.05), as well as a higher mean soft stool rate and diarrhea rate than the GT group during the dietary transition period (Table 2).

### 3.2. BCS, Fecal pH, and BW

As shown in Table 3, fecal pH in the GT group was not changed after the dietary transition period (*p* > 0.05), while the AC group had significantly reduced fecal pH (*p* < 0.05). BW fluctuated significantly after dietary change (*p* < 0.001), which is logical as the animals were growing, whereas BCS had no significant change (*p* > 0.05). In addition, there was no difference in BW between the two groups because of the same dietary intake (*p* > 0.05).

### 3.3. Fecal SCFAs and BCFAs

As shown in Table 4, fecal SCFAs had no significant change after dietary change (*p* > 0.05). Isovaleric acid was significantly reduced after GT (*p* < 0.05), while other BCFAs had no significant change after dietary change (*p* > 0.05).

### 3.4. Inflammatory Cytokines

It can be seen from Table 5 that serum inflammatory cytokines had no significant difference between the GT and AC groups after dietary change (*p* > 0.05), indicating that the current dietary change did not induce detectable inflammation in beagle puppies.

### 3.5. Fecal Microbiota Composition

As shown by anosim analysis (R > 0; Figure 2A), there was a significant difference between the AC1 and AC2 groups (*p* < 0.05), indicating that AC had a greater influence on the intestinal microbiota than GT. The α-diversity indices Chao 1, Shannon, Simpson, Ace, Goods_coverage, and Observed_species were not significantly different in the GT or AC groups before and after dietary change (Figure 2B) (*p* < 0.05).

β-diversity index analysis was performed to determine similarities between pairs of microbial communities between the two groups, and a PCoA was performed using weighted and unweighted UniFrac distance matrices. The PCoA plots showed no obvious separation of the GT or AC groups (Figure 3).

Column abundance chart analysis of the microbiota with the top 10 abundances at the phylum level testified a distinct microbiota composition among the four treatment groups (i.e., AC1, AC2, GT1, and GT2). The most abundant phyla included *Firmicutes* (49.99%), *Proteobacteria* (19.94%), *unidentified_Bacteria* (14.05%), *Fusobacteriota* (5.75%), *Actinobacteriota* (3.20%), and *Bacteroidota* (2.46%) (Figure 4A). The relative abundance of *Actinobacteriota* and Fibrobacterota tended to decrease in GT (*p* < 0.05) (Figure 4B). The column abundance chart analysis of the top 30 abundances of microbiota of the four treatment groups at the genus level is shown in Figure 4C. The most abundant genera included *Ralstonia* (18.82%), *Peptoclostridium* (9.20%), *Lactobacillus* (6.60%), *Turicibacter* (6.50%), *Allobaculum* (5.76%), and *Fusobacterium* (5.20%) in various groups. GT significantly reduced the relative abundances of *Lactobacillus* and *Clostridium_sensu_stricto_1* (*p* < 0.05) (Figure 4D). AC significantly increased the relative abundances of *Dubosiella*, while reducing those of *Clostridium_sensu_stricto_1* (*p* < 0.05). *Clostridium_sensu_stricto_1* and *Prevotella* were significantly reduced after AC (Figure 4E).

To further investigate the differential taxa abundances between the two dietary change methods, LEfSe (LDA > 4.0) was used to compare the bacterial taxa abundance in fecal samples between GT1 and GT2 and between AC1 and AC2. Ten bacteria between GT1 and GT2 (Figure 5A) and seven bacteria between AC1 and AC2 (Figure 5B) were identified. All of these identified bacteria had LDA > 4.0, alpha < 0.01 (according to the factorial Kruskal–Wallis test), indicating that the differences in the abundances of bacterial colonies had biological significance [20]. The relative abundances of *g_Turicibacter*, *s_Turicibacter_sp_h121*, *f_Ruminococcaceae*, *o_Oscillospirales*, and *g_Faecalibacterium* were high in GT2, while *s_Lactobacillus_murinus*, *o_Lactobacillales*, *g_Allobaculum*, *f_Lactobacillaceae*, and *g_Lactobacillus* were rich in abundance in GT1. Meanwhile, the relative abundances of *g_Streptococcus* and *f_Streptococcaceae* were high in AC2, and *g_Allobaculum*, *f_Peptostreptococcaceae*, *o_Peptostreptococcales_tissierellales*, *c_Clostridia*, and *s_Lactobacillus_reuteri* were abundant in AC1.

### 3.6. Fecal Metabolic Profile and Pathway Analysis

Fecal samples from the GT and AC groups were analyzed by metabolomics. After executing a series of pretreatments to correct the raw data, multivariate statistical analyses of these metabolites were performed, including PCA, PLS-DA, metabolic pathway analysis, and Spearman correlation analysis. This study analyzed the differences before and after dietary change in the GT and AC groups. The PCA score plots and PLS-DA model of GT1 and GT2 had different clusters of metabolites (Figure 6A,C). In the AC group, metabolites from AC1 and AC2 were completely separated into distinct clusters (Figure 6B,D), indicating that AC caused more dramatic changes than GT. In this study, we adopted MetaboAnalyst 5.0 to investigate the potential metabolic pathways influenced by GT and AC. Metabolites identified in the GT (*n* = 29) and AC (*n* = 50) groups were analyzed for pathways, and the charts in GT and AC are shown in Figure 6E,F. Four metabolic pathways (thiamine metabolism, aminoacyl-tRNA biosynthesis, alanine, aspartate, and glutamate metabolism, and purine metabolism) were affected after GT. Additionally, a series of metabolic pathways were affected after AC, including amino acid metabolism (i.e., aspartate, tryptophan, beta-alanine, histidine, methyl-histamine, methionine, glycine, and serine metabolism) and lipid metabolism (i.e., steroidogenesis, and the biosynthesis of phospholipid, phosphatidylcholine, and phosphatidylethanolamine).

Changes in metabolites are shown in Figure 7. Overall, AC had more metabolite changes than GT. The contents of thiamine, L-asparagine L-histidine, and citric acid were significantly upregulated, whereas those of phenylacetic acid and hypoxanthine were significantly downregulated (Figure 7A). Eight compounds (i.e., L-asparagine, L-histidine, thiamine, carnosine, histamine, citric acid, spermine, and 5-aminolevulinic acid) were significantly upregulated, while seven compounds (i.e., indoleacetic acid, serotonin, 5-hydroxyindoleacetic acid, corticosterone, phenylacetic acid, 4-pyridoxic acid, and choline) were significantly downregulated after AC (Figure 7B).

### 3.7. The Correlation Analysis of Fecal Metabolites and GM

Spearman correlation analysis was used to determine the relationship of the differential metabolites and microbiota. The results of GT are shown in Figure 8A; hypoxanthine was positively correlated with beneficial bacteria such as *Allobaculum*, *Clostridium_Sensu_Stricto_1*, and *Lactobacillus*, but negatively correlated with *Ruminococcaceae* and *Turicibacter*. Phenylacetic acid was positively correlated with *Clostridium_Sensu_Stricto_1* and *Lactobacillus*, but negatively correlated with *Faecalibacterium*. L-asparagine and thiamine were negatively correlated with *Allobaculum*, *Clostridium_Sensu_Stricto_1*, and *Lactobacillus*, but positively correlated with *Turicibacter, Holdemanella*, *Faecalibacterium*, and *Ruminococcaceae*. Hippuric acid was negatively correlated with *Lactobacillus_murinus*. In AC (Figure 8B), *Streptococcaceae* and *Streptococcus* were positively correlated with L-histidine, spermine, citric acid, carnosine, L-asparagine, and thiamine, but negatively correlated with 5-hydroxyindoleacetic acid, phenylacetic acid, corticosterone, and serotonin. In addition, *Lactobacillus_reuteri* and *Clostridium_sensu_stricto_1* had a positive correlation with choline, 5-hydroxyindoleacetic acid, phenylacetic acid, 4-pyridoxic acid, indoleacetic acid, corticosterone, and serotonin. Simultaneously, *Clostridium_sensu_stricto_1* was negatively correlated with 5-aminolevulinic acid, spermine, histamine, citric acid, L-asparagine, and thiamine. Moreover, *Allobaculum* was negatively correlated with spermine, carnosine, L-asparagine, and thiamine, but positively correlated with 5-hydroxyindoleacetic acid, phenylacetic acid, and indoleacetic acid. *Peptostreptococcaceae* and *Peptostreptococcales-tissierellales* were negatively correlated with citric acid, carnosine, L-asparagine, and thiamine, but positively correlated with 5-hydroxyindoleacetic acid.

## 4. Discussion

Dietary changes can cause alterations in pets’ bodily functions such as diarrhea, metabolic pathways, and the GM. Previous research has shown that a gradual transition can make ruminants more resilient to highly fermented diets [21,22]. In this study, we explored whether a 7-day gradual transition could alleviate the harm of dietary change on pets. Our data showed that GT significantly reduced FS, which was closer to the normal value of 2.5 points, while GT had a lower soft stool rate and did not cause diarrhea in puppies. The results indicated that, to an extent, GT could reduce the gastrointestinal responses caused by dietary change. Moreover, AC rather than GT significantly lowered fecal pH. Lin et al. found that diarrhea in beagle dogs decreased on the second day following a rapid dietary transition [23].

Although responses to dietary change can be complex, gastrointestinal symptoms are among the most observed symptoms; hence, the GM could play an important role [24]. The current study showed that dietary change also affected the GM composition.

Both AC and GT had no significant effect on the diversity of GM, but AC reduced the relative abundance of *Fibrobacterota*, which is the main bacteria that degrades fibers and is potentially involved in the production of SCFAs [25]. Both GT and AC decreased the relative abundance of *Clostridium_sensu_stricto_1*; Liu et al. found the relative abundance of *Clostridium_sensu_stricto_1* was higher in human patients with irritable bowel syndrome [26].

In the LEfSe analysis, *Turicibacter* and *Faecalibacterium* were significantly enriched after GT. It has previously been shown that low levels of *Turicibacter* can affect the gut ecosystem [27], also being associated with diseases such as depression [28]. Studies have indicated that *Faecalibacterium* can produce SCFAs, particularly butyric acid in the intestine [29], inhibit the secretion of IL-6 and IL-8 in cells, and help prevent breast cancer [30,31]. After AC, *Streptococcus* was significantly enriched, which was shown to be negatively correlated with the production of butyric acid and valeric acid [32]. It is also noteworthy that *Lactobacillus* was reduced in both methods; one explanation for this may be the different composition of the SA and CD diets, while another explanation is that the puppies were still growing. Wells found that the amount of *Lactobacillus* in the feces of weaned piglets decreased over the growing period, which could be the basis for our second explanation [33].

Beneficial bacteria such as *Peptostreptoccaceae* were reduced after AC, which may have contributed to the increase in the rate of soft stool and diarrhea in puppies. In contrast, GT increased *Ruminococcaceae* and *Faecalibacterium*. The former mainly degrades various polysaccharides and fibers to produce SCFAs [34,35], while *Faecalibacterium* has strong anti-inflammatory properties and can produce butyrate [35,36]. *Clostridium* also produces butyrate [37]. The fact that SCFAs have anti-inflammatory effects in the intestine [38,39] may help to explain why GT could alleviate soft stool and diarrhea when changing diet. Thus, as observed in the AC group, an abrupt dietary change changed the structure of the gut microbial community and resulted in intestinal discomfort in beagle puppies, but a gradual dietary transition could minimize the disturbance of the microbiota and reduce negative intestinal symptoms.

Furthermore, fecal untargeted metabolomics revealed that GT affected 4 metabolic pathway changes; surprisingly, AC affected 11 metabolic pathways, mainly including amino acid and lipid metabolisms. Both dietary change methods altered amino acid metabolic pathways, possibly due to the different dietary compositions of SA and CD diets. When compared with AC, GT caused fewer changes in metabolic pathways, indicating that GT could reduce the impact of dietary change on metabolic changes. We also found that AC significantly increased the levels of two biogenic amines, histamine and spermine. Histamine and spermine, when present in low concentrations, are considered markers of a healthy gut [40]. However, elevated levels of histamine and spermine are considered to be putrefactive and potentially carcinogenic in the gut [41,42]. Histamine is involved in the mechanism of headache from food intolerance by releasing nitric oxide from the vascular endothelium [43]. In the present study, histamine and spermine levels were significantly elevated after AC, implying that proteins were abnormally fermented in the gut, which may endanger the health of beagle puppies.

In addition, 5-hydroxyindoleacetate (5-hydroxyindoleacetic acid) declined dramatically after AC. In human studies, Shen et al. found that patients with colorectal cancer had a significant decline in fecal 5-hydroxyindoleacetate [44,45]. 5-Hydroxytryptamine (5-HT) exerts a significant role in mammalian central nervous system embryogenesis and brain ontogeny. Higher 5-HT concentrations would decrease the severity and frequency of episodes of Scottie cramp; moreover, the serum 5-HT levels of puppies affected by canine parvovirus type II increased after treatment [45,46]. Roles of 5-HT include scavenging free radicals and exerting anti-oxidative, anti-inflammatory, and analgesic effects [47,48,49]. Decreased serotonergic activity is observed in patients with neuron disorders such as depression, mania, phobia, post-traumatic stress disorder, and generalized anxiety disorder [50,51,52]. The current study showed that 5-HT declined significantly after AC. Simultaneously, 5-HT can induce the secretion of corticosterone [53]. In this study, both serotonin and corticosterone were decreased after AC. Briefly, AC raised the content of biogenic amine in the intestine, but it reduced the content of some metabolites such as 5-HT, which may have caused adverse reactions in the organism.

Next, we conducted a correlation analysis between fecal metabolites and fecal microorganisms. Results showed that hypoxanthine was positively correlated with *lactobacillus* and *Clostridium_sensu_stricto_1,* while 5-HT was positively correlated with *Clostridium_sensu_stricto_1* but negatively correlated with *Streptococcus*. Conversely, histamine and spermine were positively correlated with *Streptococcus*, but negatively correlated with *Clostridium_sensu_stricto_1* and *Allobaculum*. Collectively, changes in the population and diversity of fecal microbiota may have contributed to the alterations of fecal metabolism resulting from different dietary change methods.

In summary, this study is the first to report that GT can reduce changes in hindgut-related metabolites and the microbial community structure caused by dietary changes in beagle puppies, thereby reducing the risk of diarrhea.

## 5. Conclusions

This study comprehensively evaluated the serum and fecal change caused by different dietary change methods in beagle puppies, committed to revealing the related mechanisms underlying the metabolism and microbiota responses of GT to alleviate the adverse reactions of dietary changes. GT reduced diarrhea induced by AC throughout the dietary transition period. Moreover, the fecal microbiota was changed after different dietary changes, and the relative abundances of beneficial bacteria (i.e., *Turicibacter* and *Faecalibacterium*) in feces were increased after GT in puppies. Metabolomics analysis revealed that AC resulted in more metabolic disorders in the gut. In addition, AC increased fecal histamine and spermine levels, but decreased 5-hydroxyindoleacetic acid and serotonin levels. In summary, GT most likely attenuated diarrhea in puppies by modulating the composition and metabolism of gut microbiota. This study can be used as a basis for dietary change in pets; the physiological mechanisms involved need to be further studied.

## Figures and Tables

**Figure 1 animals-13-01300-f001:**
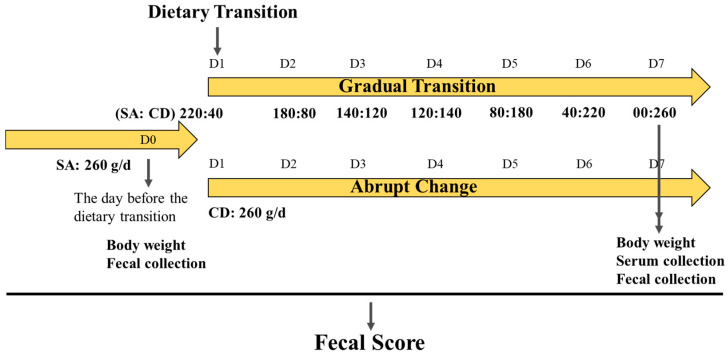
Experiment design and timeline. Beagle puppies were housed individually in metabolic cage. Body weight data and feces samples were collected on D0 and D7. Serum samples were obtained on D7. SA: salmon-based extruded diet; CD: chicken- and duck-based extruded diet. D0: the first day before the dietary transition; D1: the first day of the dietary transition; D7: the seventh day of the dietary transition.

**Figure 2 animals-13-01300-f002:**
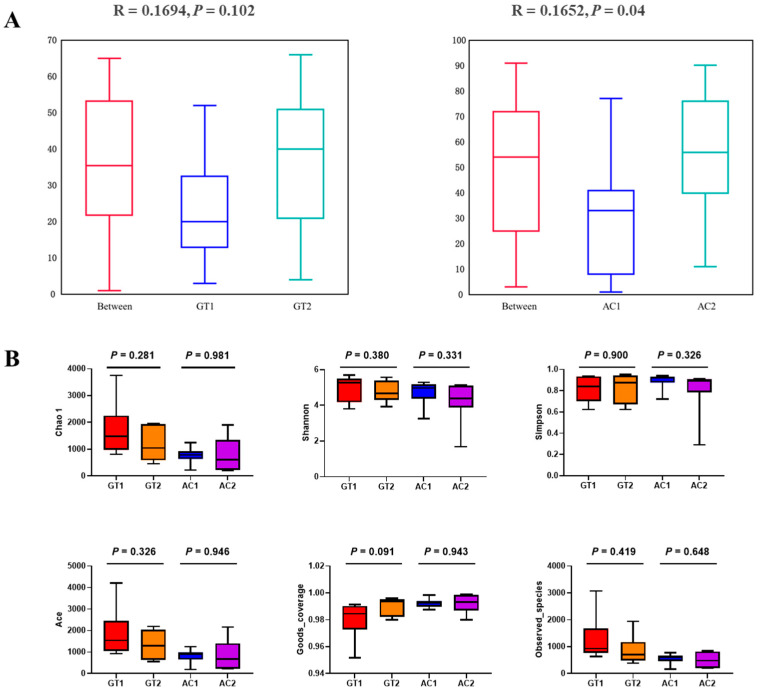
Anosim analysis and α-diversity index analysis of the GT and AC groups. The anosim test box pattern for the GT groups and AC groups (**A**), and the Chao 1, Shannon, Simpson, Ace, Goods_coverages, and Observed_species indices of the GT and AC groups (**B**). Values in (**B**) were analyzed by paired Student’s *t*-test and are presented as the means ± SE. GT1, before gradual transition, *n* = 6; GT2, after gradual transition, *n* = 6; AC1, before abrupt change, *n* = 7; AC2, after abrupt change, *n* = 7.

**Figure 3 animals-13-01300-f003:**
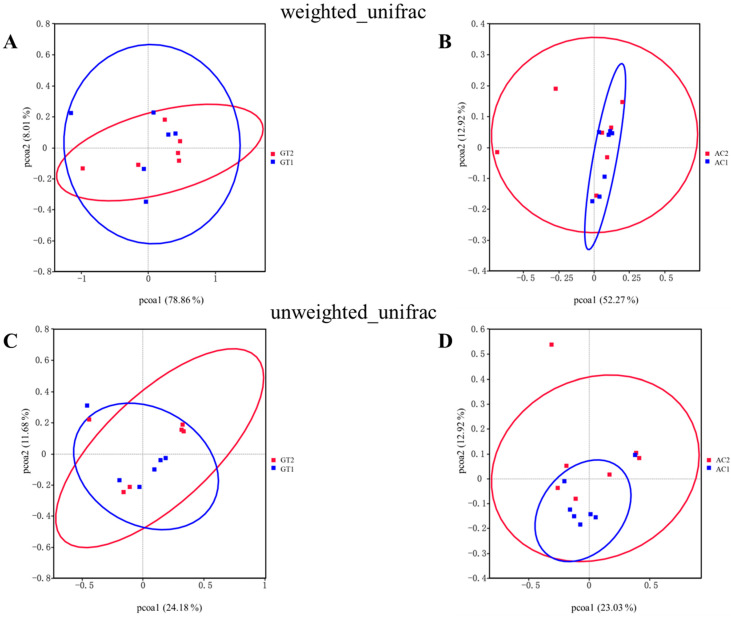
The β-diversity index analysis of GT and AC groups. The principal coordinate analysis (PCoA) of weighted_unifrac (**A**,**B**) and unweighted_unifrac (**C**,**D**) of GT groups and AC groups. GT1, before gradual transition, *n* = 6; GT2, after gradual transition, *n* = 6; AC1, before abrupt change, *n* = 7; AC2, after abrupt change, *n* = 7.

**Figure 4 animals-13-01300-f004:**
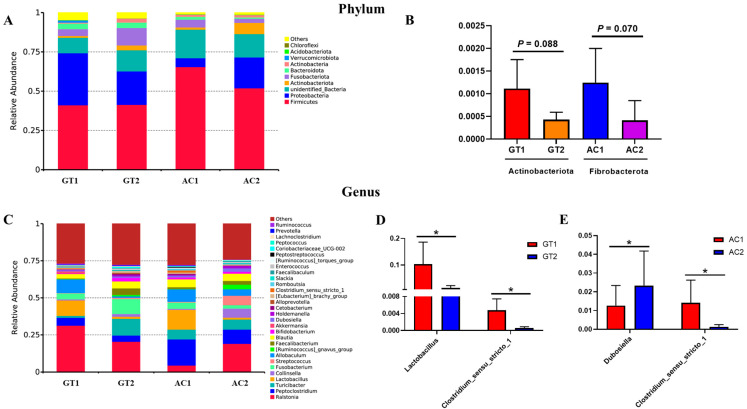
Abundance distribution and column chart of the fecal microbiota. Abundance distribution of the top 10 phyla in GT and AC groups (**A**); column chart in in GT and AC groups (**B**). Abundance distribution of the top 30 genera in GT and AC groups (**C**); column chart in GT (**D**) and AC groups (**E**). Values in (**A**–**D**) were analyzed by paired Student’s test and are presented as the means ± SE. GT1, before gradual transition, *n* = 6; GT2, after gradual transition, *n* = 6; AC1, before abrupt change, *n* = 7; AC2, after abrupt change, *n* = 7. * Significant difference between two groups (* *p* < 0.05).

**Figure 5 animals-13-01300-f005:**
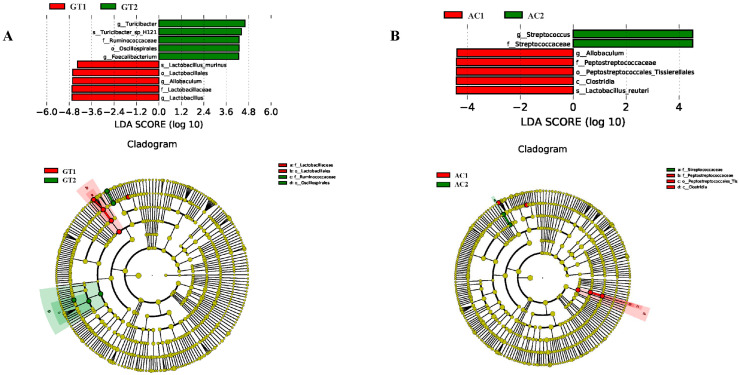
LEfSe analysis identified fecal bacteria of GT (**A**) and AC (**B**) groups. The histogram of the LDA score shows that the abundance of species differed significantly between different groups. The LDA score represents the size of the effect. In the cladogram, the circles radiating from the inside to the outside represent the classification level from the phylum to the genus (species). The diameter of each circle is proportional to the relative abundance of taxa. Red nodes refer to the bacteria that contributed greatly in GT1 or AC1, whereas green nodes refer to the bacteria dominant in GT2 or AC2. GT1, before gradual transition, *n* = 6; GT2, after gradual transition, *n* = 6; AC1, before abrupt change, *n* = 7; AC2, after abrupt change, *n* = 7.

**Figure 6 animals-13-01300-f006:**
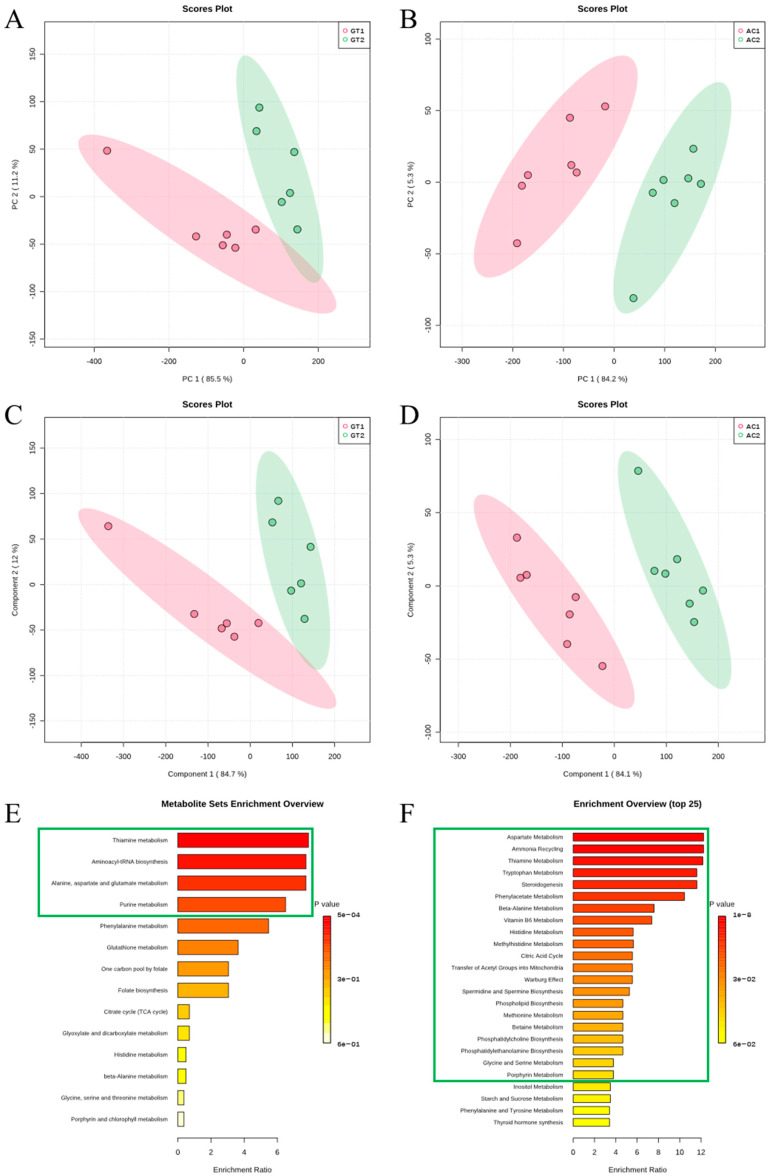
Multivariate statistical analysis of GT and AC groups. Score plots from the PCA model in GT groups (**A**) and AC groups (**B**). Score plots from the PLS-DA model in GT groups (**C**) and AC groups (**D**). Metabolism pathway analysis in GT groups (**E**) and AC groups (**F**). For (**E**,**F**), the *x* axis is the pathway impact, and the *y* axis is the pathway enrichment. Larger sizes and darker colors mean greater pathway enrichment and greater pathway impact values. GT groups, GT1 and GT2; AC groups, AC1 and AC2. GT1, before gradual transition, *n* = 6; GT2, after gradual transition, *n* = 6; AC1, before abrupt change, *n* = 7; AC2, after abrupt change, *n* = 7.

**Figure 7 animals-13-01300-f007:**
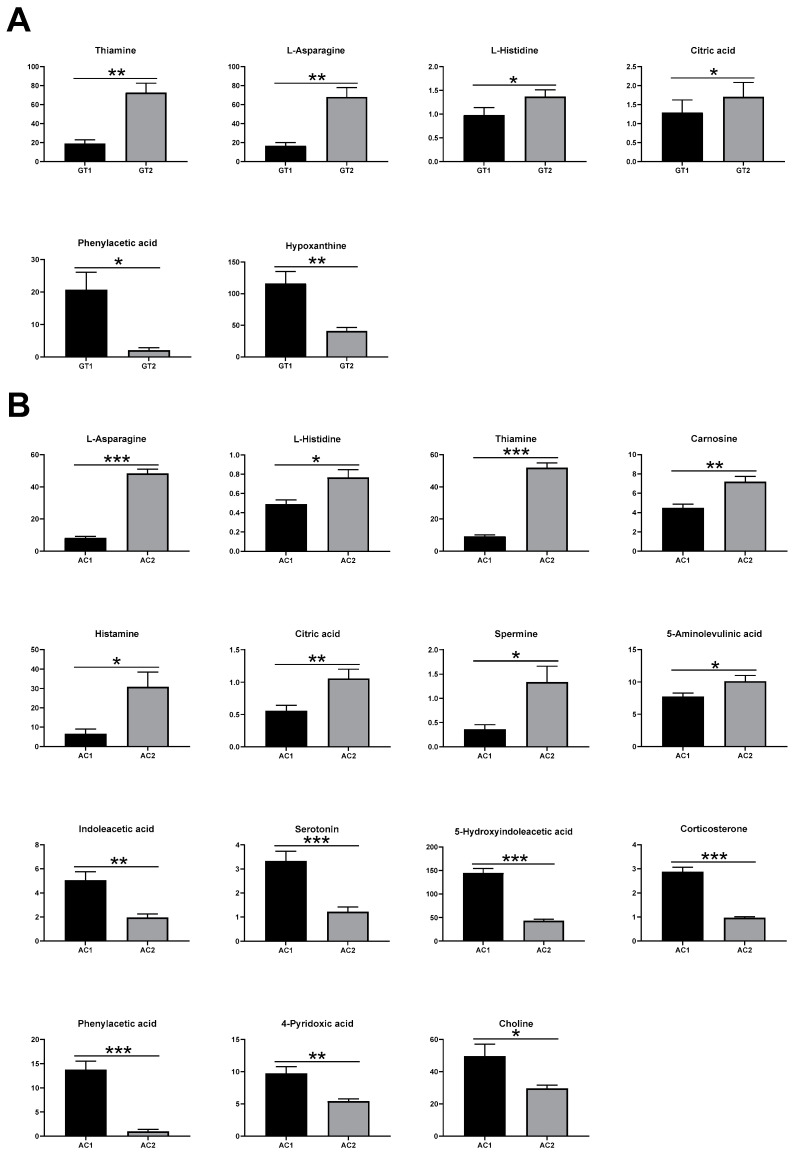
Boxplot of key metabolites of GT (**A**) and AC (**B**) groups. The *p*-values in (**A**,**B**) were analyzed by paired Student’s *t*-test and are presented as the means ± SE. GT1, before gradual transition, *n* = 6; GT2, after gradual transition, *n* = 6; AC1, before abrupt change, *n* = 7; AC2, after abrupt change, *n* = 7. * Significant difference between two groups (* *p* < 0.05, ** *p* < 0.01, and *** *p* < 0.001).

**Figure 8 animals-13-01300-f008:**
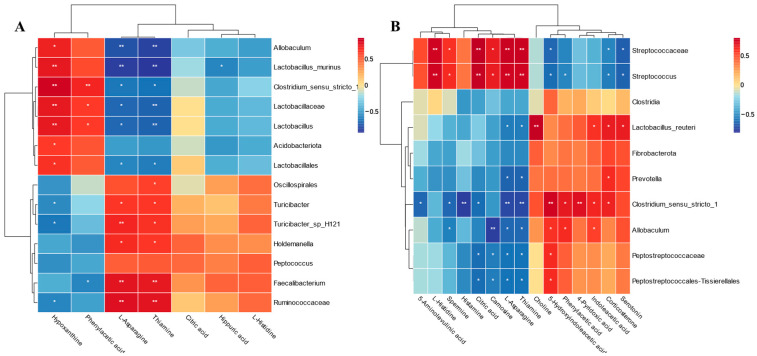
Spearman correlation analysis between the differential fecal metabolites and fecal microbiota in GT groups (**A**) and AC groups (**B**). GT groups, GT1 and GT2; AC groups, AC1 and AC2. GT1, before gradual transition, *n* = 6; GT2, after gradual transition, *n* = 6; AC1, before abrupt change, *n* = 7; AC2, after abrupt change, *n* = 7. * *p* < 0.05, ** *p* < 0.01.

**Table 1 animals-13-01300-t001:** The chemical and energy compositions of SA and CD diets.

Items *	SA	CD
DM (%)	91.19	91.68
OM (%)	91.57	92.40
CP (%)	27.69	27.91
EE (%)	11.13	11.58
TDF (%)	3.63	2.94
GE (kJ/g)	16.8	18.49

* All test methods were in accordance with the national standard. DM: dry matter; OM: organic matter; CP: crude protein; EE: ether extract; TDF: total dietary fiber; GE: gross energy. SA: salmon-based extruded diet; CD: chicken- and duck-based extruded diet.

**Table 2 animals-13-01300-t002:** Effects of different dietary change methods on FS, soft stool rate, and diarrhea rate.

Items	GT ^1^	AC ^2^	SEM	*p*-Value ^3^
FS	2.607 ^b^	2.837 ^a^	0.046	0.009
Soft stool rate	7.14%	10.20%	-	0.721
Diarrhea rate	0	10.20%	-	0.059

^1^ GT, gradual transition, *n* = 6. ^2^ AC, abrupt change, *n* = 7. ^3^ Statistical analysis of FS by unpaired Student’s *t*-test; statistical analysis of soft stool rate and diarrhea rate by chi-square test. ^a,b^ Means within a row with different superscript letters differ significantly (*p* < 0.05).

**Table 3 animals-13-01300-t003:** Effects of different dietary change methods on BCS, fecal pH, and total BW.

Groups	Items	Before Change	After Change	SEM	*p*-Value ^3^
GT ^1^	BCS	5.00	5.00	0	-
	Fecal pH	6.75	6.75	0.109	1.000
	BW (kg)	8.48 ^b^	8.85 ^a^	0.160	0.000003
AC ^2^	BCS	5.14	5.07	0.058	0.356
	Fecal pH	6.80 ^a^	6.56 ^b^	0.060	0.021
	BW (kg)	8.48 ^b^	8.86 ^a^	0.144	0.000017

^1^ GT, gradual transition, *n* = 6. ^2^ AC, abrupt change, *n* = 7. ^3^ Statistical analysis by paired Student’s *t*-test. ^a,b^ Means within a row with different superscript letters differ significantly (*p* < 0.05). BCS, body condition score; BW, body weight.

**Table 4 animals-13-01300-t004:** Fecal SCFA and BCFA contents between different groups.

Items	GT1 ^1^	GT2 ^2^	SEM	*p*-Value ^3^	AC1 ^4^	AC2	SEM	*p*-Value ^5^
Total acid (μg/g)	3919.500	4066.950	258.612	0.823	4375.781	4117.884	91.880	0.141
SCFAs (μg/g)	3390.796	3593.004	239.989	0.754	3842.939	3665.601	70.025	0.197
Acetic acid (μg/g)	1622.438	1775.754	143.217	0.677	1909.718	1809.504	31.706	0.071
Propionic acid (μg/g)	1230.793	1281.584	85.092	0.828	1370.544	1315.499	30.127	0.333
Butyric acid (μg/g)	537.565	535.666	16.698	0.967	562.678	540.597	14.607	0.518
BCFAs (μg/g)	528.704	473.946	35.865	0.226	532.841	452.284	26.642	0.110
Isobutyric acid (μg/g)	200.018	173.020	21.244	0.398	195.330	165.239	10.978	0.196
Isovaleric acid (μg/g)	289.578	256.938	15.884	0.048	290.874	239.903	14.865	0.059
Pentanoic acid (μg/g)	39.108	43.988	3.729	0.518	46.638	47.141	2.558	0.924

^1^ GT1, before gradual transition, *n* = 6. ^2^ GT2, after gradual transition, *n* = 6. ^3^ AC1, before abrupt change, *n* = 7. ^4^ AC2, after abrupt change, *n* = 7. ^5^ Statistical analysis by paired Student’s *t*-test. SCFAs, short-chain fatty acids; BCFAs, branched-chain fatty acids.

**Table 5 animals-13-01300-t005:** Inflammatory cytokine serum levels in different groups after dietary change.

Items *	GT	AC	SEM	*p*-Value
IFN-γ (pg/mL)	115.65	117.81	1.219	0.419
IL-4 (ng/L)	152.25	157.97	1.694	0.108
IL-2 (ng/L)	168.07	169.61	1.545	0.652
IL-6 (ng/L)	45.60	46.12	0.468	0.615
IgA (ng/mL)	7934.36	7993.44	65.065	0.683
IgG (μg/mL)	89.62	89.97	0.866	0.854
IgM (ng/mL)	4217.13	4238.61	39.642	0.808
TNF-α (ng/L)	81.27	81.93	0.610	0.625

* Statistical analysis by unpaired Student’s *t*-test. GT, gradual transition, *n* = 6. AC, abrupt change, *n* = 7. IFN-γ, interferon-γ; IL-4, interleukin-4; IL-2, interleukin-2; IL-6, interleukin-6; IgA, immunoglobulin A; IgG, immunoglobulin G; IgM, immunoglobulin M; TNF-α, tumor necrosis factor-α.

## Data Availability

The 16S rRNA gene sequencing data were uploaded to the National Center for Biotechnology Information at https://www.ncbi.nlm.nih.gov/bioproject/PRJNA782241 (accessed on 12 May 2022).

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
