# Peer review of "Abrupt Dietary Change and Gradual Dietary Transition Impact Diarrheal Symptoms, Fecal Fermentation Characteristics, Microbiota, and Metabolic Profile in Healthy Puppies"

_animals, 2023, doi:10.3390/ani13081300_

Round 1
Reviewer 1 Report
The authors report as purpose of the work a comparison of different dietary transition on diarrheal symptoms, fecal fermentation characteristics, microbiota, and metabolic profile in healthy dogs. The results showed that gradual transition reduced diarrhea rate in dogs most likely by modulating the composition and metabolism of gut microbiota. Although it was interesting to the readers and communities, there are several major concerns that need to be addressed.
-Are the stool scores and diarrhea rates calculated only for D7? Or do they start from D0 and continue until D7? It is not clearly indicated in the manuscript. Please, clarify.
-In the table3, the data of Fecal pH(6.808) is accurate to like 2 decimal places.
Author Response
Dear reviewer,
I have modified the manuscript according to your comments, please check.
Please see the attachment.
Kind regards,
Pinfeng Liao

Reviewer 2 Report
Both groups have microbiota composition and metabolic changes. It is assumed that the changes observed are related to the change in diet and that the difference between the 2 groups is due to the type of transition. The animals being quite young, one can also imagine that part of the changes may be linked to the natural variations of individuals. A group with no change in diet with longitudinal follow-up would have validated this hypothesis. That being said, the differences between the GT and DT groups are sufficiently marked that conclusions can still be drawn.
globally, the article is quite interesting but can still be improved. Here are some suggestions for improvement.
Phrasing should be reviewed by a language service.
32-35 The introduction is a bit naïve. The lack of food transition is already well known by veterinarians to cause diarrhea. It would be preferable to start from this known observation and to question the its origin .
41-43 why not the opposite see ref 8-11
40-45: authors consider that diet change induces development of pathogenic bacteria. It is best to refrain from saying that food transition diarrhea is due to the overgrowth of pathogenic bacteria as this statement is speculative.
46: If we respect the definition of a probiotic, Faecalibacterium is not one. Probiotic: live microorganisms that are intended to have health benefits when consumed or applied to the body
53-54: The current study investigated and compared the effects of different two dietary transition protocols, direct transition (DT) or gradual transition (GT) on dogs.
58: and further to identify potential metabolic biomarkers. It is not within the scope of this study to identify a biomarker. Avoid using this term.
Each time GM is mentioned, precise if we talk about composition or function or both
76: Clean water was freely available and twice daily food was offered (0830 and 1700)
116: Specify EP tubes
136: assay (ELISA) using commercial kits (MEIMIAN, Jiangsu, China) following the product instruction. Has the kit already been used in dogs and validated?
229: FS; mean of daily fecal score during the transition period? Or score at D7?
265 did not induce detectable inflammation in beagle dogs.
266 Inflammatory cytokine blood (or serum) levels in different groups after dietary transition.
314 of the top 10 phyla in GT and DT groups (A), column chart in in GT and DT groups (B). Abundance 314
334 togram of the LDA score shows that the abundance of species (biomarkers) differs significantly be
353 Correct capitals. Four metabolic pathways including Thiamine metabolism, Aminoacyl-tRNA biosynthesis, Alanine, aspartate and glutamate metabolism, and Purine metabolism were affected after GT.
361 Score plots from the PLS-DA model in GT groups. (C) and DT 361 groups (D). Metabolism pathway analysis in GT groups (E) and DT groups
384 thine was positively correlated with probiotics such as Allobaculum, Clostrid. The term probiotic is not suitable.
420 rephrase, meaning unclear: Although responses to dietary transition can be complex, GM could play a role and gastrointestinal symptoms are among the mostly observed [24].
422 structure of GM community in dogs=GM composition
426 The level of Clostridium_sensu_stricto_1 was higher in human patients with irritable bowel syndrome, in this study, 426 GT decreased Clostridium_sensu_stricto_1, and DT decreased Clostridium_sensu_stricto_1 427 [26].
Do you mean (relative) abundance instead of level?
432 stated that Faecalibacterium can promote produce SCFAs, particularly butyric acid in the intestine [29], inhibit the secretion of IL-6 and IL-8 in cells and help prevent breast cancer [30,31].
439 In contrast, GT increased Ruminococcaceae and Faecalibacterium, both are beneficial bacteria that can serve as probiotics. Use of the term probiotic: live microorganisms that are intended to have health benefits when consumed or applied to the body
460 In the present study, histamine and spermine levels were significantly up-regulated elevated after DT, implying that proteins were abnormally fermented in the gut, which may endanger the health of beagle dogs.
492 Moreover, fecal microbiota composition was changed after different dietary transition, and the relative abundances of probiotics (i.e., Turicibacter and Faecalibacterium) in feces were increased after GT in dogs
Author Response

(The authors gave the same response as above.)

Reviewer 3 Report
The present study is of great value for clinical veterinary prectice as , althgough a common practice, diet transitioning in diet changes benefits data are scarce. The microbiological and metabolomic data is of great value, although a cross over design might have result in more robust results.
However the manuscript have some problems that have to be adressed:
General remarks:
- Language should be intensively reviwed and checked throughout the whole document
- Parametres as soft stools and diartrea rates should be explained and statistical analises of non continuous variables
- References, mainly in the introduction, should be reviwed and idoneity checked. Data from monogastric animals is preferred.
- The title should be reviwed as the comparison is not between two transition methods but between a week period transition and an abrupt/sudden/direct (without transition) change. And should specify that the data is from PUPPIES as response and microbiome/metabolics from growing puppies are not always applied to adults.
Specific remarks
Abstract:
line 14 - consider 'GI tolerance/issu/response' rather than pet health
line 15-18 - reword the sentence and add infoprmation about treatments instead '...with the following feeding arrangement for seven consecutive days'
line 23 - probiotics is not the correct term here
Introduction:
line 35 - harmonize references
line 36 - eliminate the sentence: ' Thus, dietary transition is inevitable for pets'
line 36 - 'In other species' instead of in animals
line 36-38 - references are not related, please find references in other species sustaining the statement
line 38-39 - please find a proper reference for the statement: Sudden dietary transition often results in diarrhea in pets but the underlying mechanism remains unknown .
line 41-42: please reword: ...are considered as the main pathogens of diarrhea...
line 50: specify the specie of the statement: Recent evidence has begun to link gut microbiome and its metabolites to gastrointestinal diseases and inflammation ... humans?
M&M
line 79-80: list the ingredients with correct names: fish oil, beef meal?
line 88-89 - add the references of BCS and FS methods in the sentence : The body condition score (BCS) and fecal score (FS) of the dogs were recorded one week prior to the transition period.
and eliminate: The evaluation of BCS referred to Laflammes’ method [16] and FS referred to Middelbos’ method [17].
as is confusing
Statistical analysis - the test used for FS comparison should be reviewed
Explain how you record soft stool rate and diarrhea rate in the study and the references you use
Results
Changes in BW are logical as the animals are growing.
Discusssion
line 409 - dietary changes are not common during growth necessaraly, pleaser eliminate: 'is common during pet growth and usually' and reword: might cause alterations ....
line 417-418 - can you find monogastric data instead?
LINE 436 - can you develop further your hypothesis?
line 440-441 - probiotic is not the correct term, eliminate: both are beneficial bacteria that can serve as probiotics
line 469-471 - reference effect of serotonine in dogs and puppies instead.
please further explain whta is hour hypothesis on how 5-HT changes affect in thgis case
conclusion
line 493: avoid probiotic
please check conclusions rewording to avoid redundance
Author Response

(The authors gave the same response as above.)

Round 2
Reviewer 3 Report
Exactly as you point: for the title 'abrupt dietary change vs transition' would better explain the study
Point 4: The title should be reviwed as the comparison is not between two transition methods but between a week period transition and an abrupt/sudden/direct (without transition) change. And should specify that the data is from PUPPIES as response and microbiome/metabolics from growing puppies are not always applied to adults.
Response 4: We checked the literature and found that transition is a commonly used word for changing dietary. The following three articles can be used as examples to explain my point.
1. ‘Longitudinal fecal microbiome and metabolite data demonstrate rapid shifts and subsequent stabilization after an abrupt dietary change in healthy adult dogs’.
the reference from that point (6) remains the same:
Point 12: line 38-39 - please find a proper reference for the statement: Sudden dietary transition often results in diarrhea in pets but the underlying mechanism remains unknown.
Response 12: We have corrected the reference.